# The Mechanism of Secretion and Metabolism of Gut-Derived 5-Hydroxytryptamine

**DOI:** 10.3390/ijms22157931

**Published:** 2021-07-25

**Authors:** Ning Liu, Shiqiang Sun, Pengjie Wang, Yanan Sun, Qingjuan Hu, Xiaoyu Wang

**Affiliations:** 1Key Laboratory of Precision Nutrition and Food Quality, College of Food Science and Nutritional Engineering, China Agricultural University, Beijing 100083, China; nuli982390@163.com; 2Department of Nutrition and Health, China Agricultural University, Beijing 100193, China; Wpj1019@cau.edu.cn (P.W.); 15153515695@163.com (Y.S.); wshqlsh@163.com (Q.H.); 3Beijing Advanced Innovation Center for Food Nutrition and Human Health, China Agricultural University, Beijing 100193, China; 4Department of Gastroenterology and Hepatology, University Medical Center Groningen, University of Groningen, 9713ZG Groningen, The Netherlands; sqsun@hotmail.com; 5Department of Genetics, University Medical Center Groningen, University of Groningen, 9713ZG Groningen, The Netherlands

**Keywords:** 5-hydroxytryptamine, serotonin, secretion, metabolism

## Abstract

Serotonin, also known as 5-hydroxytryptamine (5-HT), is a metabolite of tryptophan and is reported to modulate the development and neurogenesis of the enteric nervous system, gut motility, secretion, inflammation, sensation, and epithelial development. Approximately 95% of 5-HT in the body is synthesized and secreted by enterochromaffin (EC) cells, the most common type of neuroendocrine cells in the gastrointestinal (GI) tract, through sensing signals from the intestinal lumen and the circulatory system. Gut microbiota, nutrients, and hormones are the main factors that play a vital role in regulating 5-HT secretion by EC cells. Apart from being an important neurotransmitter and a paracrine signaling molecule in the gut, gut-derived 5-HT was also shown to exert other biological functions (in autism and depression) far beyond the gut. Moreover, studies conducted on the regulation of 5-HT in the immune system demonstrated that 5-HT exerts anti-inflammatory and proinflammatory effects on the gut by binding to different receptors under intestinal inflammatory conditions. Understanding the regulatory mechanisms through which 5-HT participates in cell metabolism and physiology can provide potential therapeutic strategies for treating intestinal diseases. Herein, we review recent evidence to recapitulate the mechanisms of synthesis, secretion, regulation, and biofunction of 5-HT to improve the nutrition and health of humans.

## 1. Introduction

Serotonin, or 5-Hydroxytryptamine (5-HT), a metabolite of tryptophan (Trp), is an important gastrointestinal (GI) regulatory factor with a wide range of physiological effects on humans and animals [1,2,3,4]. Approximately 95% of 5-HT in the body is synthesized and secreted by enterochromaffin (EC) cells in the GI tract. Once 5-HT is released into the lamina propria, it is taken up by the epithelial cells through the serotonin reuptake transporter (SERT). Next, 5-HT diffuses into the bloodstream, where it is taken up by platelets and transported to peripheral target tissues. The physiological effects of 5-HT have been considerably investigated, and 5-HT has been reported to play a crucial role in GI regulation, particularly in intestinal motility and secretion [2]. The role of 5-HT in gut inflammation has also been widely investigated [4,5,6,7]. An increased concentration of 5-HT in the mucosa contributes to severe colitis. Serotonin has been shown to exert anti-inflammatory and proinflammatory effects on the gut by binding to different 5-HT receptors in animal models of inflammatory bowel disease (IBD) and colitis [8]. New clues have demonstrated that 5-HT exerts an anti-inflammatory effect on the gut by regulating the expression of the 5-HT4 receptor, with beneficial effects on intestinal epithelial cell barrier functions [4].

EC cells, which are specialized enteroendocrine (EE) cells that reside alongside the epithelium lining the lumen of the digestive tract, can synthesize and secrete 5-HT [9,10,11,12,13]. As a chemosensor, EC cells convert physiological and chemical signals from the lumen into biochemical endocrine signals through microvilli extending into the lumen and enzymes and transporters stored in the apical parts of the enterocytes. Importantly, 5-HT secretion in the gut is influenced by various factors such as nutrients, microbial community, host-derived signaling hormones, and peptides, which in turn directly or indirectly affect immune responses, nutrient metabolism, and intestinal homeostasis [14,15,16].

Gut-derived 5-HT and its various biological functions are receiving great interest from investigators. Gut-derived 5-HT possesses a range of protective effects, such as modulating gut motility and secretion, gut inflammation, liver regeneration, metabolic homeostasis, and bone remodeling, etc. This review aims to elucidate the functional role of 5-HT in and beyond the gut. We also provide an in-depth review highlighting the understanding of various factors (gut microbiota, nutrition, and hormones) in the regulation of 5-HT secretion. We hope that this review could lay a theoretical foundation for the application of 5-HT in nutrition, clinical medicine, and health.

## 2. Synthesis and Secretion of Gut-Derived 5-HT

### 2.1. 5-HT Synthesis

Only 20 of more than 700 amino acids (AAs) in nature are building blocks for proteins in cells and traditionally categorized as nutritionally essential or nonessential for humans and animals on the basis of growth or nitrogen balance [17,18]. Trp is one of nine nutritionally essential AAs [19]. In addition to its role as a substrate for protein synthesis, Trp is an important precursor for many compounds such as 5-HT, melatonin, and kynurenine [20]. Correspondingly, Trp and its metabolites play a key role in nutrition, reproduction, immune system, and anti-stress responses [21,22,23,24,25,26,27]. The kynurenine and 5-HT pathways are two main metabolic routes for Trp metabolism in mammals. Approximately 95% of the ingested Trp is degraded into kynurenine, kynurenic acid, xanthurenic acid, quinolinic acid, and picolinic acid through the kynurenine pathway. Additionally, approximately 1–2% of the ingested Trp is degraded into 5-HT and melatonin through the 5-HT pathway [28]. There are two major synthetic routes of 5-HT in the brain stem and peripheral neurons. Moreover, approximately 95% 5-HT in human body is synthesized in the peripheral system, especially in the GI tract [3,29]. Serotonergic neurons of the enteric nervous system and EC cells are two separate sources of gut-derived 5-HT in the GI tract mucosa, of which 90% 5-HT is synthesized in gut-resident EC cells, a subset of EE cells in the GI tract [30,31].

Trp hydroxylase (*TPH*), the specific serotonin-synthesizing gene, exists in two isoforms (*TPH1* and *TPH2*) [32,33]. Both *TPH1* and *TPH2* show Trp hydroxylating activity. *TPH1* is predominantly found in EC cells in the GI tract, whereas *TPH2* is mainly expressed in the central nervous system and serotonergic neurons [34]. TPH, a rate-limiting enzyme for 5-HT production, plays a key role in the conversion of Trp to 5-hydroxytryptophan (5-HTP) [1,35]. 5-HT is rapidly converted to 5-HT by aromatic L-amino acid decarboxylase (L-AADC) in the next enzymatic step [36]. Vesicular monoamine transporter 1 (VMAT1), which participates in 5-HT storage, is expressed by granules/vesicles in EC cells [37]. Newly produced 5-HT compounded with chromogranin A (CGA), an acidic protein expressed in response to 5-HT secretion, is stored in the VMAT1 vesicles of EC cells [38] (Figure 1). 5-HT stored in the dense granules/vesicles near the basal border or apical membrane of EC cells is released into the lamina propria or lumen when EC cells are exposed to intraluminal pressure or chemical and mechanical stimulation [10,15]. The biosynthesis and metabolism of gut-derived 5-HT are illustrated in Figure 1.

### 2.2. 5-HT Release and Inactivation

In serotonergic neurons, 5-HT is packaged in synaptic vesicles and then released into synapse cleft [31]. In the gut, 5-HT is mainly released from the granules stored near the basal border of the EC cell, and small amounts of 5-HT are released into lumen through the apical membrane [35]. Once released by the EC cells, there are several possible routes that 5-HT may take. 5-HT released into the lamina propria interacts with nerve terminals, epithelial cells, and immune cells or may also be taken up into the enterocytes by SERT or may enter the general circulation [39] (Figure 1).

5-HT is a positively charged molecule at physiological pH, which results in the failure of transmembrane-mediated transport. SERT relies on Na^+^ and Cl^−^ to reuptake 5-HT released from serotonergic neurons. The driving force of the reuptake process is the transmembrane ion gradient produced by Na^+^/K^+^-ATPase [40,41]. In the gut, 5-HT is also transported into surrounding enterocytes through SERT and may then be degraded into 5-hydroxyindole acetaldehyde (5-HIAL) by monoamine oxidase (MAO); 5-HIAL is, then, further transformed into 5-hydroxyindoleacetic acid (5-HIAA), which is finally excreted in urine [42,43,44]. MAO is found in the mitochondria and exists in two forms: MAO-A and MAO-B. MAO-A has a higher affinity for 5-HT [35,45,46].

5-HT released from EC cells also enter in the general circulation and are taken up by platelets via SERT. Approximately 95% of 5-HT in the blood is stored in platelets [8] in granules together with ATP, ADP, and Ca^2+^ [35]. 5-HT absorbed by platelets reaches the liver through portal circulation and is transported to peripheral target tissues through bloodstream to regulate bone density [47], liver regeneration [48,49], obesity and energy metabolism [50], and allergic airway inflammation [51]. One-third of 5-HT is converted into 5-HIAA by MAO and excreted in urine, and the remaining 5-HT is degraded into 5-HTOglucuronide through glucosidase [35].

### 2.3. 5-HT Receptors

5-HT acts in the lamina propria or lumen in a paracrine manner. In the intestinal epithelial cells or the mucosal afferent nerve of the lamina propria, 5-HT promotes intestinal motility, peristalsis, and secretion through binding to 5-HT-specific receptors (5-HTRs); 5-HTRs are classified into seven families according to structure, function, and effectiveness (5-HTR1–5-HTR7) [52,53]. Of note, the 5-HTR3 receptor is a ligand ion channel, and the other six receptors are G-protein-coupled receptors (GPCRs) [54]. In the gut, compelling evidence has shown that 5-HT regulates GI function by binding to different receptors (5-HTR1, 5-HTR2, 5-HTR3, 5-HTR4, and 5-HTR7). The conventional actions of 5-HT and its receptors in the GI tract are summarized in Table 1.

## 3. 5-HT in the Gut

EC cells are considered as “sensor cells” that have the ability to sense the luminal nutrients and non-nutrient chemicals, mechanical stimulations, and signals from the gut microbiota to release 5-HT [12,35,72]. Additionally, EC cells are stimulated to trigger the release of 5-HT in response to high intraluminal pressure changes in pH in the gut lumen [73,74,75]. Over the past decade, many studies have demonstrated that the stimulation of the intestinal cavity by gut microbiota, nutrients, and hormones could stimulate EC cells to release 5-HT. Therefore, we have described the effect of gut microbiota, nutrients, and host-derived hormones on the secretion of 5-HT in greater details.

### 3.1. Gut Microbiota and 5-HT Release

Gut microbiota are a complex and dynamic population of microorganisms that inhabit the GI tract of humans and other mammals [76]. Over the past decade, gut microbiota have received considerable attention because of their functional role in regulating host physiology, metabolism, and immunity [77,78]. Emerging evidence has also shown that the gut microbiota play a critical role in regulating host 5-HT secretion in EC cells by interacting with various compounds produced by the host or gut microorganisms [76]. Short-chain fatty acids (SCFAs), as markers of bacterial metabolism [79], enhance colonic *TPH1* mRNA expression by interacting with EC cells [16]. This finding is consistent with previous research that intraluminal administration of SCFAs into the proximal colon significantly augments the release and production of 5-HT by accelerating colonic transit through stimulating 5-HTR3, and thereby, promoting colonic contraction [80]. In contrast, 5-HT production following the stimulation of EC cells by SCFAs via triggering the entry of extracellular Ca^2+^ is unchanged [14], which indicates that the interaction between microorganisms and the host plays an indispensable role in 5-HT secretion. A study conducted by Yano et al. demonstrated that microbial-specific metabolites such as SCFAs, α-tocopherol, tyramine, and p-aminobenzoate promote TPH1 expression and 5-HT release [81]. These results suggest an association between gut microbiota communities and host in regulating the basic biological processes through 5-HT [81]. Many different types of GPCR sensors of microbial metabolites are expressed in colonic EC cells, including olfactory receptor 558 (Olfr558), free fatty acid receptor 2 (FFAR2), olfactory receptor 78 (OLF78) that senses SCFAs, G-protein-coupled receptor 35 (GPR35) that senses small aromatic acids, G-protein-coupled bile acid receptor 1 (GPBAR1) that senses secondary bile acids, and G-protein-coupled receptor 132 (GPR132) that senses lactate and acyl amides. These receptors are activated in the process of 5-HT secretion by various gut microbial metabolites [10,13].

Most 5-HT is produced by EC cells, and a small amount of 5-HT is synthesized by a deconjugation process of glucuronide-conjugated 5-HT by a bacterial enzyme such as β-glucuronidase [12]. The gut microorganisms metabolize various substances through their interaction with the host, thereby affecting the release of 5-HT. The elucidation of the gut microbiome and host genetics in the past 10 years has helped to clarify the relationship between gut microbiota and the physiological and pathological conditions of the host. Consequently, the mechanism of microbial dependence that affects the physiological function of the host is likely to be elucidated, which would be beneficial to find methods for using gut microbial intervention to improve body health. Jonathan et al. reported that *Escherichia coli* Nissle 1917, one of the currently available probiotic bacteria, regulates THP1 through the interaction between host and probiotics by enhancing 5-HT level and its bioavailability in ileal tissues [42]. Because of the complexity of the gut microflora, beneficial bacteria promote body health through the 5-HT system, while pathogenic bacteria may cause intestinal diseases by damaging the intake of 5-HT. Enteropathogenic *E. coli*, a foodborne pathogen, inhibits SERT activity by reducing protein tyrosine phosphatase, and the damaged SERT function is associated with infectious diarrheal diseases [82]. Many studies have confirmed that the gut microbial flora and its particular metabolites influence the biosynthesis of 5-HT. However, it is largely unknown whether the alteration of 5-HT level caused by host–microbiota interaction in turn affects the colonization, growth, or adaptation of enteric microorganisms. Therefore, much work is required to investigate the metabolic pathway and molecular mechanisms of microbial metabolites in regulating 5-HT levels in the gut.

### 3.2. Nutrients and 5-HT Release

The specialized EE cells are dispersed as single cells scattered throughout the epithelium of the GI tract from the stomach to the rectum and are considered as the largest endocrine system of the human body [83]. EE cells regulate various physiological and homeostatic functions both within and outside the gut by secreting various hormones and peptides [84]. EC cells represent around 50% of all EE cells that sense diverse dietary nutrients and metabolites to produce ~95% of total body 5-HT [14]. Studies conducted on human primary colonic EC cells and BON cells (immortalized cell line models of EC cells) found that 5-HT is released from EC cells in response to stimulation of luminal nutrients [11,14,15,85]. Unlike human primary colonic EC cells, BON cells release 5-HT following the stimulation of luminal D-glucose through sodium-glucose-linked transporter 1 (SGLT1) [86]. Additionally, the sensing of glucose in the lumen is related to the expression of SGLT3 in EC cells, which results in the release of 5-HT [87,88]. Ingested food components are digested by digestive enzymes into a form that can be absorbed into the bloodstream. Glucose is the main form of carbohydrate absorbed by mammals and serves as a luminal substance to trigger several key events in the physiological regulation of the intestinal tract [89].

As a chemosensor in the GI mucosa, EC cells release 5-HT by sensing the presence of glucose, thereby inhibiting gastric emptying and food intake by activating 5-HTR3 on exogenous afferent nerves of rodents and humans [60,90]. Intriguingly, the nutrient sensing capacity of EC cells in 5-HT secretion from the mouse duodenum and colon is region-specific. Carbohydrate absorption is generally achieved over the entire small intestine, and only a small amount of glucose reaches the colon. Correspondingly, EC cells in the colon are more sensitive to glucose than those in the duodenum [14], as glucose transporter 1 (GLUT1) is highly expressed in colonic EC cells, while glucose transporter 2 (GLUT2) is abundantly expressed in duodenal EC cells [11]. The low glucose availability leads to the upregulation of GLUT1, which is a high-affinity and low-capacity glucose transporter. On the other hand, GLUT2, a low-affinity and high-capacity glucose transporter, is upregulated in a high concentration of glucose [91].

Zelkas et al. reported that 5-HT-secreting EC cells show enormous diversity in response to acute and chronic changes in glucose availability [92]. Acute exposure to high concentration of glucose results in 5-HT release from EC cells, which involves the entry of Ca^2+^ and an increment in the number of vesicles for exocytosis. Chronic exposure to fasting-related levels of glucose leads to the enhancement of 5-HT synthesis through transcriptional regulation of TPH1. Consistently, food deprivation enhances gut-derived 5-HT synthesis accompanied by enhancement of lipolysis in adipocytes and liver gluconeogenesis, as well as prevention of glucose uptake in hepatocytes [93]. The enhancement of 5-HT synthesis in response to the elevated level of luminal glucose after feeding promotes gut motility and peristaltic reflex through the activation of ascending and descending interneurons to facilitate digestion. 5-HT also plays a pivotal role in enhancing body fat degradation and liver gluconeogenesis during fasting, which contributes to the maintenance of the blood glucose level.

### 3.3. Hormonal Control of 5-HT Release

The enteroendocrine system is responsible for secreting a diverse range of gut hormones, which play a highly important role in the physiological regulation of the GI tract [84]. EC cells coexist closely with other EE cells, instead of existing in “one cell type” solitarily along the length of the GI tract. A recent discovery is that EE cells communicate with EC cells locally through paracrine action in the gut [94]. The glucagon-like peptide 1 (GLP-1) receptor is particularly highly expressed in EC cells. The neighboring GLP-1-storing EE cells secrete GLP-1, and GLP-1 then stimulates EC cells to release 5-HT through the activation of GLP-1 receptors [13]. The GLP-1 receptor agonist has been reported to release 5-HT in both small intestine and colon. Of note, the spontaneous secretion of 5-HT was higher in the duodenum when compared with that in the colon. However, a significant enhancement in 5-HT release was detected with the treatment of a GLP-1 receptor agonist both in the duodenum and colon [13]. Moreover, it has been reported that 5-HT enhances nutrient-induced GLP-1 release from ileal segments through a process involving interactions with 5-HT receptors [95]. Using the intestinal secretin tumor cell line (STC-1) for further exploration in vitro, results revealed that 5-HT (30 or 100 μM) significantly enhanced GLP-1 secretion in STC-1 cells when compared with control group. Additionally, the non-specific 5-HT receptor antagonist asenapine inhibited the 5-HT-promoted GLP-1 release, which supports the 5-HT receptor-mediated mechanism [95]. Further studies are needed to investigate the interaction and mechanisms between the secretion of 5-HT and GLP-1. EC cells are also sensitive to endogenous regulatory molecules. Norepinephrine-mediated stimulation of EC cells activates alpha-2A adrenergic (Adrα2A) receptors through catecholamines, which leads to chronic visceral hypersensitivity [10]. Hormone crosstalk exists between gut mucosal EC cells and the neighboring enterocytes within the epithelium, but its complex effects remain unknown.

## 4. Physiological and Pathophysiological Role of Gut-Derived 5-HT

### 4.1. 5-HT and Gut Inflammation

Accumulating evidence through clinical and animal studies indicates that 5-HT, as a signaling molecule in the intestine, plays a pivotal role in intestinal inflammation (Figure 2). 5-HT signaling has been investigated in an animal model of intestinal inflammation, including 2,4,6-trinitrobenzene sulfonic acid (TNBS)-induced colitis [96] and ileitis [5,97,98,99], dextran sodium sulfate (DSS)-induced colitis [4,100], and trichinella spiralis infection-induced intestinal inflammation [101]. Several studies have revealed that 5-HT is a key proinflammatory signaling molecule in gut inflammation because of the enhanced concentration of intestinal 5-HT and downregulation of SERT expression under intestinal inflammation [6,65,102,103]. Because of the knockout of *TPH1*, the concentration of 5-HT in the GI tract was significantly reduced, followed by alleviation in clinical severity and histological damage scores by pharmacological adjustment of mucosal 5-HT in DSS- or dinitrobenzene sulfonic acid (DNBS)-induced colitis [104]. Consistently, several studies have reported that a THP1 or TPH inhibitor alleviates the severity of colitis and plays a protective role in colitis [105,106]. Additionally, SERT transcription is reduced during intestinal inflammation, which contributes to impaired absorption of 5-HT [2,107].

5-HT plays an important role in the proinflammatory or anti-inflammatory process through binding to different receptors [5,64]. The anti-inflammatory role of 5-HT is accompanied by the activation of epithelial 5-HTR1A and 5-HTR4. Compared to control, the severity of experimental colitis in mice was enhanced through intraluminal administration of a 5-HTR1A and 5-HTR4 antagonist [4,64]. Upregulation of 5-HTR4 expression protects the large intestine from DSS- or TNBS-induced colitis by maintaining epithelial integrity, stimulating the proliferation of crypt epithelial cells, and reducing apoptosis [64]. A study reported that treatment with a 5-HTR2A antagonist (Ketanserin) alleviates intestinal inflammation by improving gut integrity, reducing the production of inflammatory cytokines in macrophages, and inhibiting the activation of nuclear factor-κB (NF-κB) in experimental colitis; this result further confirmed the deleterious role of 5-HTR2A on intestinal inflammation [108]. However, the 5-HT receptors involved in the proinflammatory and anti-inflammatory processes reported in the current literature are contradictory. Spohn et al. demonstrated that chemical activation of 5-HTR4 reduced the severity of TNBS- and DSS-induced colitis [64]. In contrast, Rapalli et al. found that the inhibition of 5-HTR4 improves the progression and pathological outcome of TNBS-induced colitis, thus suggesting the detrimental effect of 5-HTR4 on TNBS-induced colitis [5]. Kim et al. also reported that the inhibition of 5-HTR7 signaling reversed acute and chronic colitis induced by DSS or TNBS [109]. In contrast, several research studies have demonstrated that the development of colitis was not affected by 5-HTR7 [4,5]. Thus, further studies are required to determine the role of 5-HT receptors on experimental colitis to reveal the association between 5-HT receptors and the downstream signaling pathways under inflammatory conditions (Figure 2).

The immune response to inflammation involves the extensive proliferation of immune cells and aberrant production of immune mediators and cytokines such as tumor necrosis factor (TNF)-α, interferon (IFN)-γ, interleukin (IL)-1β, IL-6, and IL-8 and their related signaling pathways [110,111]. 5-HT receptors have been identified in human and rodent immune cells [44]. EC cells are in close proximity with immune cells in the gut mucosa, suggesting the existence of interaction between EC cells and immune cells [112]. Immune cells, including dendritic cells, macrophages, neutrophils, lymphocytes, and B lymphocytes, proliferate in the 5-HT-mediated proinflammatory response [4,113], suggesting that 5-HT plays a vital part in the immune response. Recent studies have shown that 5-HT signaling is altered by proinflammatory cytokines such as TNF-α, IL-1β, IL-6, and IFN-γ, as well as the anti-inflammatory cytokine IL-10 by regulating the expression and function of SERT. Intriguingly, several studies have found that IFN-γ, TNF-α, and IL-6 and a low concentration of IL-10 caused a significant decrease in the function and activity of epithelial SERT [4,114,115,116].

### 4.2. 5-HT and Liver Regeneration

5-HT acting on the liver is entirely derived from the gut because of the lack of 5-HT synthesis capacity in hepatocytes [117]. 5-HT activated in platelets is released in the liver and mediates liver regeneration after partial hepatectomy and inhibits liver regeneration in the *TPH1* gene knockout mice [48,49]. Liver regeneration was mediated by promoting DNA synthesis and cell proliferation through acting on 5-HTR2 [118] and 5-HTR7 [119,120]. Another study revealed that SERT knockout in platelets has no effect on liver regeneration, thus indicating that the extremely low level of 5-HT in plasma is sufficient for liver regeneration [121].

### 4.3. 5-HT and Energy Homeostasis

Metabolic homeostasis is regulated by nerves and hormones. Several recent studies have shown that 5-HT, an important endocrine substance and hormone, regulates the metabolic function of many tissues and influences obesity and energy metabolism [1,50,122,123]. The liver, a pivotal organ in an organism’s metabolism, plays a central role in regulating plasma glucose metabolism and energy metabolism [124]. 5-HT cannot be produced by hepatocytes; hence, all the peripheral 5-HT in the liver is derived from the gut. A previous study revealed that 5-HT produced during fasting promotes gluconeogenesis by enhancing the activity of two key gluconeogenesis rate-limiting enzymes (glucose 6-phosphatase and fructose 1,6-bisphosphatase) through 5-HTR2B [93]. The cyclic AMP that is the downstream of 5-HTR2B is enhanced at transcriptional level after the elevated activity of two key enzymes; subsequently, cAMP-dependent protein kinase A (PKA) and CREB are activated [125]. Additionally, gut-derived 5-HT in hepatocytes prevents glucose uptake in a GLUT2-dependent manner, thereby further favoring the maintenance of blood glucose levels [93]. Because *TPH1* is expressed in adipocytes, the regulation of 5-HT in adipose tissue is more complicated than that in the liver. TPH1-produced 5-HT in adipocytes regulates the metabolism of adipose tissue through local autocrine signals [50,126,127]. In white adipocytes, 5-HT synthesized in EC cells enhances the phosphorylation and activity of hormone-sensitive lipase (HSL) through binding to the 5-HT2B receptor, therefore elevating circulatory free fatty acids and glycerol [93]. There are two possible pathways to promote lipolysis and inhibit lipogenesis: (1) HSL is activated indirectly by cAMP and cAMP-dependent protein kinase A (PKA); (2) perilipin is phosphorylated by PKA and, consequently, stimulates phosphorylation of HSL [128]. Enhanced glycerol acts as a fuel of gluconeogenesis and is converted into acetyl-CoA by β-oxidation for the synthesis of ketone bodies [1]. Because of the complexity of the serotonergic system in adipose tissue, more studies are required to elucidate the underlying responsible role for 5-HT in the future.

### 4.4. 5-HT and Bone Remodeling

5-HT and its role in bone metabolism are receiving great interest from researchers. Bone remodeling and renewal is a highly integrated process, which includes bone resorption through osteoclasts and bone formation through osteoblasts. These two processes are dynamically balanced, which contributes to the maintenance of bone [129]. Low-density lipoprotein receptor-related protein-5 (Lrp5) is essential for Wnt signaling to form bones [130,131,132]. Previous studies have reported that Lrp5 is expressed in osteoblasts and EC cells in the GI tract [133]. However, Lrp5 could act in EC cells in the gut, not in osteoblast, to regulate bone-mass accrual via a Wnt-independent pathway [134]. Lrp5 inhibits the expression of TPH1, thereby reducing 5-HT concentration in the blood. Less 5-HT binds to 5-HTR1B in osteoblasts and 5-HTR1B signaling is reduced in osteoblasts. As a result, the expression and function of cyclic AMP response element binding protein (CREB) is enhanced, which promotes cyclin expression and results in enhanced osteoblasts differentiation and proliferation [134]. In this process, 5-HT derived from the GI tract and transported through the circulation is detrimental to bone formation through inhibiting osteoblast proliferation [134] (Figure 3). Consistently, some studies have supported that gut-derived 5-HT could suppress bone growth in rats [135,136]. Thus, pharmacological inhibition of gut-derived 5-HT synthesis through the inhibitor of THP1 may be a potential bone anabolic treatment for low bone mass [137,138]. Additionally, there are conflicting results in the model where Lrp5 regulates bone mass through duodenal 5-HT. A study conducted by Cui et al. demonstrated that gut-derived 5-HT synthesis is not associated with Lrp5 [131]. Growing evidence has shown that 5-HT plays an important role in bone metabolism. However, because of the different synthesis sites of 5-HT, including brain-derived 5-HT [139], gut-derived 5-HT [134,137,140], and bone-derived 5-HT [141], 5-HT has different roles in bone metabolism (Figure 3).

## 5. Conclusions

5-HT synthesized in EC cells has been recognized for decades as an important signaling molecule in the gut. It is well known that 5-HT derived from neurons and EC cells is involved in the regulation of GI peristalsis, sensation, and secretion. Approximately 95% of 5-HT in the body is synthesized and secreted by EC cells in the GI tract. The findings of several studies have suggested that gut microbiota, nutrients, and hormones could stimulate EC cells to release 5-HT. New clues from recent studies expand our understanding of the functional role of gut-derived 5-HT in and far beyond the gut. As an important neurotransmitter and hormone in the GI tract, research on 5-HT is increasing, but the underlying mechanisms of the relationship between 5-HT and physiological actions in the body remain largely unclear. Therefore, it is essential to highlight the functional role of 5-HT and various factors (gut microbiota, nutrients, and hormones) in the regulation of 5-HT secretion in order to facilitate the application for 5-HT in nutrition, clinical medicine, and health.

## Figures and Tables

**Figure 1 ijms-22-07931-f001:**
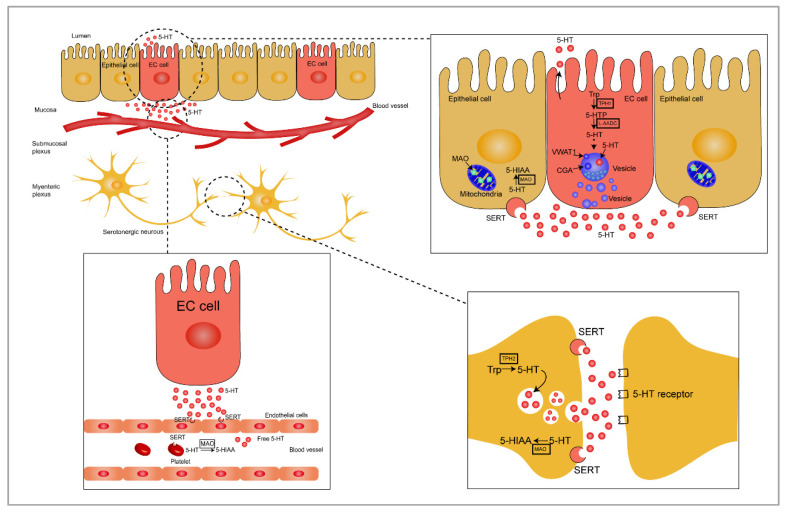
Schematic representation of gut-derived 5-HT biosynthesis and metabolism. Enterochromaffin (EC) cells and serotonergic neurons convert tryptophan into 5-HTP through the rate-limiting enzymes, TPH1 and TPH2, respectively, and the newly formed 5-HTP is rapidly degraded into 5-HT by L-AADC. The synthesized 5-HT and CGA are rapidly packaged into vesicles through VMAT1. EC cells express sensory receptors by acting as chemosensors to continuously release 5-HT in response to stimuli in the luminal environment, including chemical and mechanical stimulation, luminal pressure, and nutritional and intestinal microbial metabolites and hormones. Most of the 5-HT is released into the extracellular space from the bottom of EC cells, and a comparatively smaller amount of 5-HT is released into the lumen through the apical membrane. The surrounding enterocytes take up 5-HT by SERT, and 5-HT is then metabolized to 5-HIAA through MAO in the mitochondria. As a neurotransmitter, the synthetized 5-HT is packaged in synaptic vesicles in serotonergic neurons and released into the synapse cleft. Serotonin exerts a high effect on the postsynaptic membrane through the 5-HT receptors and reabsorbs on the presynaptic membrane through SERT. SERT is also detected in endothelial cells and platelets where 5-HT entering into the lamina propria is taken up and then converted into 5-HIAA or transported to peripheral target tissues. Trp, tryptophan; 5-HT, 5-hydroxytryptamine; TPH1, tryptophan hydroxylase 1; TPH2, tryptophan hydroxylase 2; 5-HTP, 5-hydroxytryptophan; 5-HIAA, 5-hydroxyindoleacetic acid; L-AADC, L-amino acid decarboxylase; CGA, chromogranin A; VMAT1, vesicular monoamine transporter 1; MAO, monoamine oxidase; SERT, serotonin reuptake transporter.

**Figure 2 ijms-22-07931-f002:**
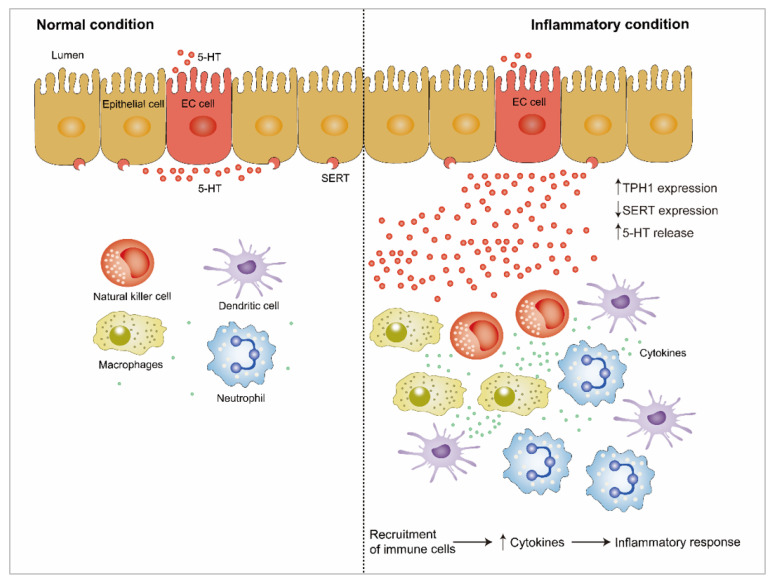
The role of gut-derived 5-HT under inflammatory condition and inflammatory condition. THP1 and SERT expression, as well as 5-HT release can be altered under inflammatory condition. Enhanced 5-HT promotes recruitment of immune cells, such as natural killer cells, dendritic cell, macrophages, and neutrophil during inflammation. Subsequently, enhanced cytokines production is released from immune cells, which can promote inflammatory response. Upwards pointing arrows indicate an enhancement, and downwards pointing arrows indicate a decrease. 5-HT, 5-hydroxytryptamine; TPH1, tryptophan hydroxylase 1; SERT, serotonin reuptake transporter.

**Figure 3 ijms-22-07931-f003:**
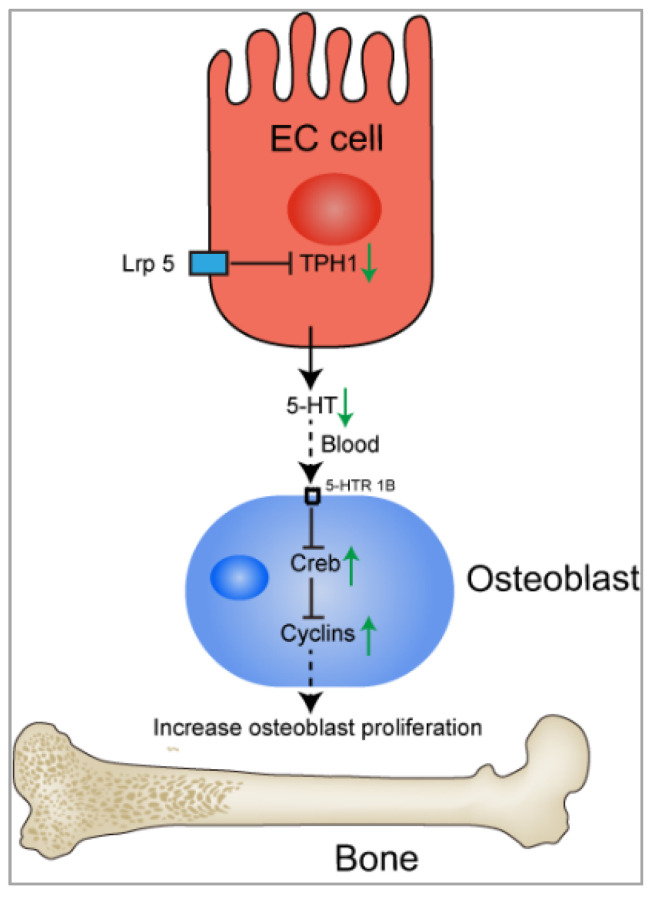
The action of Lrp5 and 5-HT on the regulation of bone formation. Lrp5 inhibits the expression of TPH1 in cells. As a result, reduced 5-HT concentration in circulation reduces 5-HTR1B signaling in osteoblasts. Cyclic AMP response element binding protein (CREB) and cyclin expression is enhanced, which favors osteoblasts proliferation and bone formation. Lrp5, low-density lipoprotein receptor-related protein-5; 5-HT, 5-hydroxytryptamine; CREB, cyclic AMP response element binding protein.

**Table 1 ijms-22-07931-t001:** Conventional effects of 5-HT and its receptors in the gastrointestinal tract.

Conventional Effect	Pathway	Mediated Receptors	References
Motility and peristaltic reflex	Activate ascending and descending interneurons	5-HT3 and 5-HT4 receptor	[2,55,56,57]
Secretion (bicarbonate and electrolyte)	Neural mediated or through paracrine pathway acts on nearby enterocytes	5-HT2, 5-HT3 and 5-HT4 receptor	[58,59]
Pancreatic secretion and gastric emptying	Activate vago-vagal reflex and act in synergy with cholecystokinin (CCK)	5-HT2 and 5-HT3 receptor	[60,61,62]
Vasodilation	Locally regulate blood vessel diameter through intrinsic reflex circuits	5-HT3 and 5-HT4 receptor	[2,63]
Inflammation	The pro-inflammatory actions by promoting an inflammatory offensive to protect the gut from invasion and the anti-inflammatory actions by inducing neurogenesis	5-HT1A, 2A, 2B, 2C, 5-HT3, 5-HT4, and 5-HT7 receptor	[4,5,6,64,65,66]
Neurogenesis and enteric protection	Play an important role though Neuronal 5-HT	5-HT4 receptor	[67,68,69]
Mucosal growth	Serotonergic neurons project submucosal cholinergic neurons	5-HT2A receptor	[70,71]

## Data Availability

Not applicable.

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
