# Peer review of "The Mechanism of Secretion and Metabolism of Gut-Derived 5-Hydroxytryptamine"

_ijms, 2021, doi:10.3390/ijms22157931_

Round 1

Reviewer 1 Report

I read with interest the manuscript entitled "The mechanism of secretion and metabolism of gut-derived 5-hydroxytryptamine". However, some comments can be made on some points of the manuscript :

1- In table 2, line 3, only 5-HT3 is mentioned for pancreatic secretion and gastric emptying, but it has been shown that the 5-HT2 receptor is also involved (Li et al., 20O1)

2- The authors did not report the close proximity of enterochromaffin cells and immune cells in the gut, which explains that one can modulate the function of the other

3- It is not superfluous to mention the factors that are capable of stimulating  EC cells release 5-HT, namely neuronal stimulation acetylcholine, high intraluminal pressure and low pH (Bulbring & Lin 1958, Spiller 2008).

4- Since inflammation of the intestines is a major public health problem, it would have been useful to draw a figure on which would be represented on one side the roles of 5-HT under normal conditions and on the other the roles of 5-HT under inflammatory conditions.

5- Considering the pandemic proportion of obesity in the world, is there any idea about the link between 5-HT and gut inflammation and implication with obesity?

6- The authors state: "5-HT synthesis in response to elevated luminal glucose after feeding promotes intestinal motility and the peristaltic reflex by activating ascending and descending neurons to facilitate digestion". However, deletion of TPH1 (TPH1KO) , which eliminates 5-HT from enterochromaffin cells, does not affect constitutive gastrointestinal motility. how can the authors explain these observations?

7- Do the authors know whether GLP-1 releases 5-HT in the small intestine or in the colon, and if so, does this release occur by different or similar mechanisms ?

8- The interaction between 5-HT and GLP-1 deserves to be further developed by the authors.

Author Response

Comments and Suggestions for Authors

I read with interest the manuscript entitled "The mechanism of secretion and metabolism of gut-derived 5-hydroxytryptamine". However, some comments can be made on some points of the manuscript:

Reply:

Thanks for reviewing our manuscript and the constructive comments. Your comments and suggestions are of great value for our study. Our point-by-point responses are shown below and heighted in revised version.

  1. In table 2, line 3, only 5-HT3 is mentioned for pancreatic secretion and gastric emptying, but it has been shown that the 5-HT2 receptor is also involved (Li et al., 2001)

Reply:

Sorry for our mistake. According to for your suggestions, we added 5-HT2 receptor in the table 1, line 3 based on the citated paper.

  1. The authors did not report the close proximity of enterochromaffin cells and immune cells in the gut, which explains that one can modulate the function of the other

Reply:

We are sorry for missing the very important information. We added the information of the close proximity between EC cells and immune cells in the gut mucosa, please see details in line 269-270.

  1. It is not superfluous to mention the factors that are capable of stimulating EC cells release 5-HT, namely neuronal stimulation acetylcholine, high intraluminal pressure and low pH (Bulbring & Lin 1958, Spiller 2008).

Reply:

Thanks a lot for your suggestions. High intraluminal pressure and the changes of pH in the gut lumen have been reported to induce the release of 5-HT from EC cells. We added those information as you suggested. Thanks again!

Ginzel & Kottegoda (1954) suggested that 5-HT stimulated baroreceptors as well as chemoreceptors in the carotid sinus; and since the action of acetylcholine, when applied in low concentrations to the mucosal surface of the intestine, resembled that of 5-HT, the same may be true for the small intestine. 5-HT has been shown to increase the release of both acetylcholine and calcitonin gene-relate d peptide from stimulated submucosal intrinsic primary afferent neurones (IPANs).

  1. Since inflammation of the intestines is a major public health problem, it would have been useful to draw a figure on which would be represented on one side the roles of 5-HT under normal conditions and on the other the roles of 5-HT under inflammatory conditions.

Reply:

Thanks for pointing out this. Because of your useful question. we drew a figure (Figure 3) for roles of 5-HT under normal conditions or inflammatory conditions.

  1. Considering the pandemic proportion of obesity in the world, is there any idea about the link between 5-HT and gut inflammation and implication with obesity?

Reply:

Thanks a lot for your suggestion. Gut inflammation has been associated with elevated mucosal 5-HT content (British Journal of Pharmacology. 2018 May;175(9):1535-1547; American Journal of Physiology-Gastrointestinal and Liver Physiology. 2003 Jul;285(1): G207-16). Previous study reported that augmented circulatory and gut-derived 5-HT was observed in obese humans (International Journal of Obesity. 2018 Nov;42(11):1880-1889), thereby indicating that 5-HT is likely to drive obesity in humans. 5-HT plays an important role in bridging gut inflammation and obesity, however, “cause-effected” relationship between gut inflammation and obesity remains largely elusive. Gut inflammation associated with obesity by effect of 5-HT was further confirmed.

  1. The authors state: "5-HT synthesis in response to elevated luminal glucose after feeding promotes intestinal motility and the peristaltic reflex by activating ascending and descending neurons to facilitate digestion". However, deletion of TPH1 (TPH1KO), which eliminates 5-HT from enterochromaffin cells, does not affect constitutive gastrointestinal motility. how can the authors explain these observations?

Reply:

Thanks for pointing out this. Deletion of TPH1 did not affect 5-HT biosynthesis from enteric nerves (Science. 2003 Jan 3;299(5603):76). It has been reported that constitutive gastrointestinal motility was unchanged in TPH1 deficient mice (Journal of Neuroscience. 2011 Jun 15;31(24):8998-9009), suggesting neuronal 5-HT are more contributory to gut motility than mucosal 5-HT. However, Heredia et al. demonstrated that mucosal 5-HT is essential for peristaltic reflexes (The Journal of Physiology. 2013 Dec 1;591(23):5939-57).

  1. Do the authors know whether GLP-1 releases 5-HT in the small intestine or in the colon, and if so, does this release occur by different or similar mechanisms?

Reply:

Thanks a lot for your question. We added the information of release of 5-HT by GLP-1 in the small intestine and colon in the revised version.

  1. The interaction between 5-HT and GLP-1 deserves to be further developed by the authors.

Reply:

Thanks a lot for your suggestions. We added the further interaction between 5-HT and GLP-1 in the manuscript.

GLP-1 receptor agonist has been reported to release 5-HT in both small intestine and colon. Of note, the spontaneous secretion of 5-HT was higher in the duodenum when compared with that in the colon. However, a significant enhancement in 5-HT release was detected with the treatment of GLP-1 receptor agonist both in the duodenum and colon (Lund et al. 2018). Moreover, it has been reported that 5-HT enhances nutrient-induced GLP-1 release from ileal segments through a process involving interactions with 5-HT receptors (Ripken et al. 2016). Using the intestinal secretin tumor cell line (STC-1) for further exploration in vitro, results revealed that serotonin (30 or 100 μM) significantly enhanced GLP-1 secretion in STC-1 cells when compared with control group. Additionally, non-specific serotonin receptor antagonist asenapine inhibited the serotonin-promoted GLP-1 release, which supports the 5-HT receptor-mediated mechanism (Ripken et al. 2016). Further studies are needed to investigate the interaction and mechanisms between the secretion of 5-HT and GLP-1.

Reviewer 2 Report

In the last five years (2016 to 2021), 193 reviews were found in the PubMed using the query “5-HT AND gut” what makes very short the space for an original approach to the topic.
In the present format, the authors failed to explain what remains to be elucidate about “the functional role of 5-HT in and beyond the gut” and how the present manuscript addresses the issues that remain to be elucidated. 
The organization of the manuscript needs to be imporved. Point 2.2 is more about 5-HT inactivation than about 5-HT release and could include the hepatic metabolism (presented in 3.2) in lines 156 – 158.
The section about the physiological roles of gut-derived 5-HT present interesting information. However, organization of this section needs to be extensively reviewed. The section about 5-HT and bone remodeling is not clear. After all, what is the role of Lrp5 with 5-HT signaling? Does it make sense to present the statement about the relevance of 5-HT in bone metabolism at the end of the respective section? The topic is about gut 5-HT, but it is not clear if there is a special contribution of the gut 5-HT to the bone metabolism. 
The section about the 5-HT and energy homeostasis is not clear. The authors should improve the discussion about the liver and adipocyte mechanisms. The study mentioned in the text (line 193) is not quoted and is questionable the strategy to circumvent a more consistent explanation about the effects of 5-HT in metabolism with the “excuse” of the complexity of the topic (lines 203 – 204).
The section about 5-HT and gut inflammation also deserves a significant improvement. Are we talking about pro- or anti-inflammatory effects because the type of 5-HT receptors are different?
The section about “Gut microbiota” (4.1) repeats information already presented and the section “Nutrients” (4.2) presents interesting data about certain nutrients may induce gut release of 5-HT but this formation is scattered in the text and loses relevance.

In summary, a review about the particular mechanisms of 5-HT formation, its release and effects in the gut and in other organs that may contribute to a better knowledge of the link between gut/neuronal axis and the link between gut 5-HT and metabolism is interesting and useful for a more comprehensive knowledge of the effects of gut 5-HT. However, the present version seems to be very preliminary and deserve an extensive improvement. 

Author Response

Comments and Suggestions for Authors

  1. In the last five years (2016 to 2021), 193 reviews were found in the PubMed using the query “5-HT AND gut” what makes very short the space for an original approach to the topic.

Replay:

Thanks for your comments. 5-HT has been recognized for decades as an important signaling molecule in the gut. Indeed, many reviews were published to elucidate the importance of tryptophan and its metabolites. Research on 5-HT is increasing, but the underlying mechanisms of the relationship between 5-HT and physiological actions in the body remain largely unclear. This review highlighted the functional role of 5-HT and various factors (gut microbiota, nutrients, and hormones) in the regulation of 5-HT secretion. The present paper will enable a better understanding of 5-HT in disease prevention and treatment in order to facilitate the application for 5-HT in nutrition, clinical medicine, and health.

  1. In the present format, the authors failed to explain what remains to be elucidate about “the functional role of 5-HT in and beyond the gut” and how the present manuscript addresses the issues that remain to be elucidated.

Reply:

We are sorry for the unclear statement. As an important neurotransmitter and hormone in the GI tract, research on 5-HT is increasing, but the underlying mechanisms of the relationship between 5-HT and physiological actions in the body remain largely unclear. Additionally, recent studies reported that 5-HT plays a key role in liver regeneration, bone remodeling, energy homeostasis, gut inflammation, and so on. These published data indicate the functional role of 5-HT in and beyond the gut. This paper just highlighted the functional role of 5-HT and various factors (gut microbiota, nutrients, and hormones) in the regulation of 5-HT secretion in order to facilitate the application for 5-HT in nutrition, clinical medicine, and health.

  1. The organization of the manuscript needs to be imporved. Point 2.2 is more about 5-HT inactivation than about 5-HT release and could include the hepatic metabolism (presented in 3.2) in lines 156 – 158.

Reply:

We are sorry for the unclear statement. 5-HT release was briefly described in point 2.2, more information on 5-HT release was presented in Figure 1, please see details in Figure 1.

  1. The section about the physiological roles of gut-derived 5-HT present interesting information. However, organization of this section needs to be extensively reviewed. The section about 5-HT and bone remodeling is not clear. After all, what is the role of Lrp5 with 5-HT signaling?

Reply:

Thanks a lot for your comments. Lrp5 acts in gut EC cells, not in osteoblasts, to control bone formation via a Wnt-independent pathway. Gut-derived 5-HT binds to 5-HTR1B and inhibits cyclic AMP response element binding protein (CREB) expression and function in osteoblast. Subsequently, the expression of cyclin is downregulated, leading to reduced osteoblast proliferation. However, Lrp5 could diminish the action of 5-HT. More information was added in this section.

  1. Does it make sense to present the statement about the relevance of 5-HT in bone metabolism at the end of the respective section? The topic is about gut 5-HT, but it is not clear if there is a special contribution of the gut 5-HT to the bone metabolism.

Reply:

Thanks a lot for your suggestions. Although 5-HT acts as signaling molecule in brain and periphery. 5-HT derived from the GI tract and transported through the circulation is the major source for skeletal 5-HT (Cell. 2008 Nov 28;135(5):825-37). Gut-derived 5-HT inhibits bone formation, and Lrp5 plays an important role in this pathway. However, little is known about how Lrp5 affects TPH1 expression in EC cells. Lrp5 inhibits the expression of TPH1 in cells. As a result, reduced 5-HT concentration in circulation reduces 5-HTR1B signaling in osteoblasts. Cyclic AMP response element binding protein (CREB) and cyclin expression are enhanced, which favors osteoblasts proliferation and bone formation. The action of Lrp5 and 5-HT on the regulation of bone formation were presented in Figure 2.

  1. The section about the 5-HT and energy homeostasis is not clear. The authors should improve the discussion about the liver and adipocyte mechanisms. The study mentioned in the text (line 193) is not quoted and is questionable the strategy to circumvent a more consistent explanation about the effects of 5-HT in metabolism with the “excuse” of the complexity of the topic (lines 203 – 204).

Reply:

Thanks a lot for your suggestions. We have improved the discussion about the liver and adipocyte mechanisms of gut-derived 5-HT. Additionally, we added citation in line 193 and rephrased this section,please see details in Point 3.4.

  1. The section about 5-HT and gut inflammation also deserves a significant improvement. Are we talking about pro- or anti-inflammatory effects because the type of 5-HT receptors are different?

Reply:

Thanks a lot for your suggestion. The regulatory role of 5-HT is associated with the proinflammatory receptors and anti-inflammatory receptors of 5-HT under inflammatory conditions. It has reported that 5-HTR1A and 5-HTR4 are anti-inflammatory receptors (The Journal of Nutrition. 2020 Jul 1;150(7):1966-1976; Gastroenterology. 2016 Nov;151(5):933-944.e3) and 5-HTR2A is proinflammatory receptors (International Journal of Molecular Medicine. 2016 Mar;37(3):659-68). Additionally, we drew a figure (Figure 3) for roles of 5-HT under normal conditions or inflammatory conditions.

  1. The section about “Gut microbiota” (4.1) repeats information already presented, and the section “Nutrients” (4.2) presents interesting data about certain nutrients may induce gut release of 5-HT but this formation is scattered in the text and loses relevance.

Reply:

Thanks a lot for your suggestion. We read through this paper and did not find similar information with “Gut microbiota” section.

We reorganized those sentences in “Nutrients” section to avoid scattered and low relevance. Please see details in those sections.

  1. In summary, a review about the particular mechanisms of 5-HT formation, its release and effects in the gut and in other organs that may contribute to a better knowledge of the link between gut/neuronal axis and the link between gut 5-HT and metabolism is interesting and useful for a more comprehensive knowledge of the effects of gut 5-HT. However, the present version seems to be very preliminary and deserve an extensive improvement.

Reply:

Thanks for your constructive comments. Tryptophan is a rate-limiting essential amino acid and thus a building block of life. Current research on the physiology and pathophysiology of tryptophan metabolism has revealed the central role of tryptophan and its metabolites. The discovery of a broad range of bioactive compounds derived from tryptophan will enable a better understanding of the unique role of this amino acid in disease prevention and treatment. The present study herein highlights the functional role of 5-HT and various factors (gut microbiota, nutrients, and hormones) in the regulation of 5-HT secretion in order to facilitate the application for 5-HT in nutrition, clinical medicine, and health. This will demonstrate the relevance of tryptophan and its metabolites in nutrition and health.

Round 2

Reviewer 2 Report

The authors were very polite in their response. However they only addressed the questions raised, minimally. 

In my view, a review to update the gut-derived 5-HT is interesting. However, the organization of the manuscript is not optimal and hard for the reader to follow and to catch the main messages. In my view, without a major change in the organization of the manuscript, the message is hard to catch and the interest of the manuscript is lost. 

To be more specific, I recommend the following sequence:

  1. Introduction (making sure the last paragraph gives more focus to the gut-derived 5-HT).
  2. Synthesis and secretion of gut-derived 5-HT
    1. 5-HT synthesis
    2. 5-HT release
    3. 5-HT receptors
    4. Inactivation (presenting MAO and uptake mechanisms) and hepatic metabolism.
  3. 5-HT in the gut
    1. Regulation of 5-HT secretion
    2. Gut microbiota
    3. Nutrients and 5-HT release
    4. Platelets and 5-HT 
    5. Hormonal control of 5-HT release
  4. Physiological and pathophysiological role of gut-derived 5-HT
    1. 5-HT and gut inflammation
    2. 5-HT and liver regeneration
    3. 5-Ht and energy homeostasis
    4. 5-Ht and bone remodeling
  5. Conclusion

Author Response

In my view, a review to update the gut-derived 5-HT is interesting. However, the organization of the manuscript is not optimal and hard for the reader to follow and to catch the main messages. In my view, without a major change in the organization of the manuscript, the message is hard to catch and the interest of the manuscript is lost. 

To be more specific, I recommend the following sequence:

  1. Introduction (making sure the last paragraph gives more focus to the gut-derived 5-HT).
  2. Synthesis and secretion of gut-derived 5-HT
    1. 5-HT synthesis
    2. 5-HT release
    3. 5-HT receptors
    4. Inactivation (presenting MAO and uptake mechanisms) and hepatic metabolism.
  3. 5-HT in the gut
    1. Regulation of 5-HT secretion
    2. Gut microbiota
    3. Nutrients and 5-HT release
    4. Platelets and 5-HT 
    5. Hormonal control of 5-HT release
  4. Physiological and pathophysiological role of gut-derived 5-HT
    1. 5-HT and gut inflammation
    2. 5-HT and liver regeneration
    3. 5-Ht and energy homeostasis
    4. 5-Ht and bone remodeling
  5. Conclusion

Reply: Thanks for reviewing our manuscript and the constructive comments. We provided a major change in the organization of the manuscript to make manuscript is  optimal and easy for the reader.

Round 3

Reviewer 2 Report

The new organization make the manuscript easier to follow and focus the text in the main topic: the gut-derived 5-HT. In my view, some changes should still be introduced as suggested in the attached file.

Author Response

The new organization make the manuscript easier to follow and focus the text in the main topic: the gut-derived 5-HT. In my view, some changes should still be introduced as suggested in the attached file.

Reply:

Thaks for your hard work! We have changed the organization according to your constructive comments. We appreciate the expertion of you in 5-HT related reseach.